# Physical drivers and trends of the recent delayed withdrawal of the Southwest Monsoon over Mainland Indochina

Kyaw Than Oo [1, 2, *], Chen Haishan[1,*], Kazora Jonah[1,3], Du Xinguan[1]

[1] Key Laboratory of Meteorological Disaster, Ministry of Education/Joint International Research Laboratory of Climate and Environment Change/Collaborative Innovation Center on Forecast and Evaluation of Meteorological Disasters, Nanjing University of Information Science and Technology, Nanjing, 210044, People's Republic of China

[2] Aviation Weather Services, Yangon, Myanmar

[3] Rwanda Meteorology Agency, Kigali, Rwanda

*Corresponding author: haishan@nuist.edu.cn ,kyawthanoo34@outlook.com

**Key point**

- Cumulative Change of Mainland Indochina Southwest Monsoon (MSwM) new definition index improves understanding of monsoon transitions.
- Anomalous trends of Subtropical Westerly Jet and Tropical Easterly Jet are linked to changes in wind patterns and monsoon timing.
- Anomalous Sea surface temperatures impact moisture transport during MSwM Retreat phases.

**Plain Language Summary**

The study investigates the delay withdrawal of the Mainland Indochina Southwest Monsoon (MSwM) by using spatial trend connections with meteorological and oceanic factors. The new Cumulative Change-Point Monsoon (CPM) definition index well described the definition of monsoon seasonal shifting. The results show that the subtropical westerly jet is getting stronger while the tropical easterly jet is getting weaker within these years. This influences the regional wind patterns and delays the monsoon withdrawal. The study highlights the critical role of ocean-atmosphere interactions and local atmospheric circulation in influencing the summer monsoon. Specifically, warmer sea surface temperatures in the Indian Ocean enhance moisture transport through strengthened southwesterly winds, while atmospheric pressure gradients drive moisture convergence over the region. These processes contribute to prolonged monsoon seasons, increasing the risk of floods and disrupting agricultural schedules, which significantly impact water management and farming in Mainland Indochina.

## Abstract

The study investigates the key factors that cause the Mainland Indochina Southwest Monsoon (MSwM) to delay withdrawal, utilizing a spatial trend correlation between the monsoon index and various meteorological and oceanic variables such as sea surface temperature (SST), zonal winds, and moisture transport. A significant strengthening trend in the Subtropical Westerly jet (SWJ) and a weakening Tropical Easterly jet (TEJ) not only impacts regional wind patterns but also delays the monsoon departure. The anomalous South China Sea and the equatorial Indo-Pacific Ocean surface temperature (SSTA) further contribute to these delayed withdrawals, and there is a significant correlation between the MSwM withdrawal index and SSTA, moisture transport, and essential atmospheric factors. The results clarify MSwM dynamics, offering significant insights for future climate research associated with MSwM. The study also suggests that the variability of ocean-atmosphere interactions and local atmospheric circulation patterns is critical for understanding monsoon variability, which has a potential impact on climate predictions, water resource management, and agriculture practices over Mainland Indochina.

**Keywords**: Mainland Indochina, Monsoon Withdrawal, MSwM, SWJ, TEJ, ENSO

## 1    Introduction

In tropical Asia, the summer monsoon system is one of the most significant meteorological phenomena in the Northern Hemisphere. This monsoon onset and withdrawal are the most notable intraseasonal variable in monsoon systems. The beginning of the summer rainy season, extensive convection, and a rapid change in atmospheric circulation characterize this period (Aung et al., 2017; Bordoni and Schneider, 2008; Salinger et al., 2014). Based on previous science literature of the Asia-Pacific monsoon classification, there are three primary types of summer monsoons, East Asian , Indian, and Western North Pacific  monsoons (Wang and Ho, 2002),( Supplementary Fig S-1). The eastern bay of  Bengal (EBOB), as known as the mainland-Indochina region (MIC) study area (Fig. 1a) is situated in a transitional zone between the ISM (India Summer Monsoon) and the WNPSM (Western North Pacific Summer Monsoon) systems (Oo and Jonah, 2024). The monsoon indices had been developed to study the transition and boundary between the Indian Summer Monsoon (ISM) and East Asian Summer Monsoon (EASM) (Cao et al., 2012), characterize monsoon onset and withdrawal using rainfall-based metrics (Bombardi et al., 2019; Zhang et al., 2024), and define these phases through circulation-based approaches (Chen et al., 2023; Hu et al., 2022). The MIC also features complex terrain, with high mountain ranges and long costal area. Simply, the MIC dominates a unique position between the southern areas of East and Middle East Asia, where this monsoon system over MIC exhibiting transitional characteristics between the two monsoon systems (Zhang et al., 2002a).

Consequently, significant variation in agricultural planting and ploughing times occur over MIC
affected by the monsoon rainfall (Fig. 1b), depending upon the early or late monsoon onset or
withdrawal.

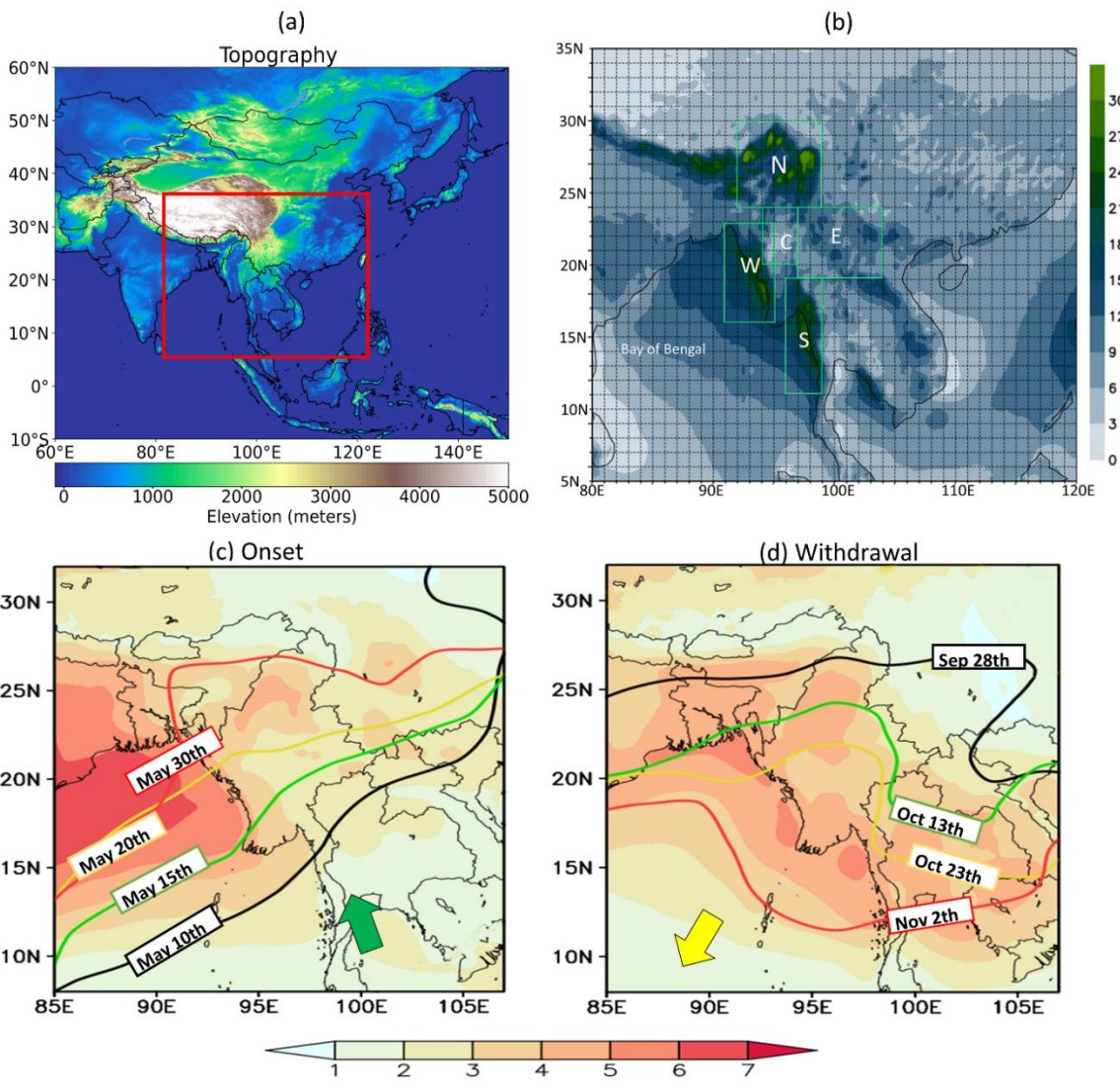

*Fig. 1 (a) Topography (m) of the study area, including mainland Indochina. (b) Daily rainfall (mm) during the MSwM season. (c) Climatological onset and (d) withdrawal dates of MSwM with standard deviation values (shaded, days). This figure was created with Python 3.10 (Matplotlib 3.5.2 [https://matplotlib.org/], Cartopy 0.20.0 [https://pypi.org/project/Cartopy/]).*

A range of onset and withdrawal indices has been established, based on rapid changes in extensive
atmospheric structures. Especially, the most commonly used atmospheric variables for defining onset
and withdrawal indices include rainfall (Ajayamohan et al., 2009; Colbert et al., 2015; Htway and
Matsumoto, 2011; Vijaya Kumari et al., 2018), and reversable component wind (CY Li, 1999; Li et
al., 2010; Webster and Yang, 1992). In addition to precipitation and circulation, the thermal and
moisture characteristics of the atmosphere also serve as an important indicator for describing the
progression of the monsoon season (Song et al., 2025; Zhang et al., 2012). The summer monsoon
typically onset to MIC between mid-May and early June, with slight variations in indices and statistics

(Mao and Wu, 2007; Oo, 2023a; Ren et al., 2022; Wang and Ho, 2002). The MSwM withdrawal
displayed significant interannual variability, with a extent of one to two weeks may vary among the
earliest and latest withdrawals based on climatological data (Evan and Camargo, 2011; Oo, 2023a). In
addition to ENSO, recent studies have demonstrated that mid-high latitude systems also have
significant impacts on ENSO, East Asian monsoon onset and withdrawal, which should also be briefly
reviewed (Hu et al., 2020, 2025)

The global wind circulation and the El Niño Southern Oscillation (ENSO) have been widely
studied for their influence on the interannual variability of monsoon onset (Roxy et al., 2014; Wu,
2017), the formation of South Asia's subtropical high (Q Guo, 1988; Wang et al., 2008; Zhang et al.,
2002b), and fluctuations in local sea surface temperature (SST) (Salinger et al., 2014; Xu et al., 2023).
Based on these long-term physical atmospheric variables data, this study seeks to examine the factors
contributing to the delay withdrawal of the MSwM, with superior weight on ocean-atmosphere
interactions and zonal wind dynamics, which have been insufficiently explored in this area, since
monsoon rains are crucial for agriculture and fill up water supplies (Win Zin and Rutten, 2017; Zin
Mie Mie Sein et al., 2015). In this study, we present the variability of withdrawal dates over interannual
scale. Due to the significant up trending of local withdrawal date of MSwM, derived from the
combination of reversal of winds circulation (Ramage, 1971) and vertical moisture flux transport
changes (Fasullo and Webster, 2003). We investigate the mechanism driver of these delay withdrawal
and potential driver of continues untimely rainfall after MSwM withdrawal.

## 2   Data and Method

The study utilizes data from five sources:

1. ***Department of Meteorology and Hydrology, Myanmar (DMH)***: Daily observed rainfall, sea
level pressure, and annual onset and withdrawal dates for significant regions were collected from
DMH, which operates 79 meteorological stations nationwide. This data help assess validate of
reanalysis datasets.

2. ***NCEP/NCAR Reanalysis***: This dataset provides zonal (u) and meridional (v) wind components,
specific humidity (q), geopotential height (z), and vertical velocity (w) at atmospheric isobaric
levels in the troposphere for wind analysis (Kanamitsu et al., 2002).

3. ***European Centre for Medium-Range Weather Forecasts - ECMWF***: ERA5 offers reanalysis
data with a 0.25° geographical resolution for global climate analysis, including sea level pressure
(SLP), moisture flux convergence (MFC), and outgoing longwave radiation (OLR) for the period
from 1991 to 2020 (Hersbach et al., 2020).

4. *Unified Gauge-Based Analysis of Global Daily Precipitation (CPC)*: This dataset provides rainfall data (Chen et al., 2008; Jiao et al., 2021).

5. *Hadley Centre*: Hadley Centre Sea Surface Temperature dataset (HadISST) (Selman and Misra, 2014).

**2.1.1   Definition of Monsoon onset and withdrawal by CPM index**

The MSwM region is defined by coordinates 10°N–30°N and 85°E–110°E (Fig. 1; see Appendix for additional details). We examine seasonal fluctuations in the moisture budget and extensive atmospheric circulation, as established by:

*Equation 1*

$$\mathbf{MFC} = -\int_{Surface}^{300hPa} \boldsymbol{\nabla}_p \cdot (\boldsymbol{U}q) \frac{dp}{g} = \mathrm{P} - \mathrm{E} + \frac{\partial W}{\partial t}$$

This equation was developed from a prior study on the variability of the Asian Monsoon (Walker et al., 2015). In this context, Moisture Flux Convergence (MFC) is a vital quantity that delineates the equilibrium of moisture in the atmosphere. The initial segment of the equation encapsulates the dynamic component, represented by the divergence of moisture flow "**(Uq).∇ p**" denotes the movement and accumulation of moisture resulting from wind patterns. This dynamic element is essential for comprehending how atmospheric circulation patterns affect moisture availability. The second component, "**P - E + ∂W/ ∂t**" signifies the thermodynamic equilibrium of moisture inside the system. **P** represents precipitation, **E** signifies evaporation, and ∂W/∂t reflects temporal variations in water storage. This relationship illustrates how thermodynamic mechanisms regulate the moisture budget and influence the overall climate dynamics of the monsoon zone.

By integrating dynamic and thermodynamic aspects, the cumulative change of the MSwM (CPM) index provides a strong framework for analyzing the behavior of the monsoon circulation over time. Building upon the MFC, we define the Cumulative Change of the MSwM (CPM) index of onset and withdrawal as follows:

*Equation 2*

$$\mathbf{MSwM\ (CPM)} = \frac{1}{5} * (\mathbf{D(U_1 - U_2) + D(P_1 - P_2) + D(MFC) + D(TP_{net}) + D(OLR))}$$

These five diagnostic variables were used to characterize the onset and withdrawal of the monsoon over the transitional Mainland Indochina (MIC) region, outgoing longwave radiation (OLR), vertically integrated moisture flux convergence (MFC), net precipitation (TpNet), meridional shear in zonal winds (U1–U2), and pressure gradient (P1–P2). These variables are physically consistent with the governing moisture budget equation. We determine the normalized values for each factor annually for statistical investigation. The cumulative value change from positive to negative, or vice versa, is

verified for further statistical calculations. "D" in Equation (2) expresses the date when the state shifts of positive or negative (+ to - or - to +) values and typically represents the change or difference in the standardize values of each variable in a year.

Thus, changes in MFC directly link large-scale circulation dynamics with rainfall variability, while TpNet (P–E) and OLR confirm convective activity and cloud cover. In this equation, D (U1-U2) and D(P1 - P2) represent the differences in zonal winds and pressure between the southern and northern regions of the Mainland Indochina (MIC). Specifically, the southern region (90°–100°E, 10°–15°N) reflects the influence of the broad Indochina Peninsula, where the southwest monsoon winds are most active, while the northern region (95°–100°E, 25°–30°N) captures the terrain-influenced pressure dynamics near the eastern Tibetan Plateau (Fig. 1a) and the southwest monsoon wind withdrawal pattern (Fig. 1d), we take pressure readings that are different from longitude ranges as two distinct regions. The meridional shear in the 850-hPa zonal winds and the pressure gradient between northern and southern regions which is driving monsoon flows, the key indicators of monsoon circulation, are averaged across two distinct regions: the southern MIC (90E-100E, 10N-15N), referred to as (U1,P1), and the northern MIC (95E-100E, 25N-30N), designated as (U2,P2). This approach follows the Gill-type tropical circulation response (Gill, 1980), where deep convection excites westward-propagating Rossby wave responses that enhance southwesterlies to the west of the convention center, and the South China Sea–Bay of Bengal circulation system provides a dynamical link between ISM and WNPSM (Wang et al., 2009; Wang and Zhou, 2024). Consequently, the five indices together capture the coupled thermodynamic and dynamic drivers of monsoon evolution in this transitional region. The term D(MFC) captures the cumulative changes in moisture transport and convergence, essential for monsoon rainfall, while D(TPnet) represents net precipitation changes, indicating monsoon withdrawal as well as onset by rainfall and D(OLR) the changes in outgoing longwave radiation, closely linked to convective activity and cloud cover to confirm monsoon rainfall, respectively. We calculate the mean change date of the standardized positive/negative value of the outgoing longwave radiation (OLR), the vertically integrated moisture budget transition (MFC), the net precipitation (TpNet), the meridional shear wind (U1-U2) (U-wind), and the pressure differential (P1-P2) (dP). The first day of three consecutive positive or negative days is taken into consideration when determining the change date. Next, we obtained each variable's change point dates for every year. Lastly, the climatology data for every term date was acquired (Supplementary Table S1). We used these findings to compute the MSwM Change Point Index, which is the arithmetic mean onset dates, withdrawal dates, and season length (Supplementary Table S4). A student's t-test is used to calculate the correlation coefficients of these findings at the 95% level of significance. This rounded approach allows for a comprehensive assessment of the interrelationships among these parameters, simplifying

the identification of key onset (Fig. 1c) and withdrawal (Fig. 1d). Moreover, common statistical methods such as correlation (Krugman et al., 2018), regression (Ma, 2019), random forest (Breiman, 2001), box and whisker (Schmidhammer, 2000) are also applied in the study at necessary parts.

The Random Forest technique, a widely used ensemble machine-learning method, was utilized to find the relative relevance of variables controlling monsoon withdrawal and rainfall. It generates several decision trees during training by sampling subsets of data and features, hence mitigating overfitting and enhancing generalization (Breiman, 2001). Our study incorporated input variables comprising atmospheric and hydrological factors, including Outgoing Longwave Radiation (OLR), Net Precipitation (Net), Moisture Flux Convergence (MFC), Zonal Wind Shear (U), and Pressure Differential (dP). Each tree generated a prediction, and the final output was ascertained by averaging (for regression tasks) or by majority voting (for classification tasks). Box and whisker plots were employed to graphically encapsulate the distributions of essential variables across various phases of the monsoon season (Schmidhammer, 2000). It is good to examine the day-of-year distributions for monsoon withdrawal timing based on many factors, including dP, U, MFC, Net Precipitation, and OLR. This analysis clearly exhibited variability and key tendencies in the data, highlighting the contribution of specific variables to withdrawal patterns. For example, zonal wind shear (U) exhibits narrower variability, indicating a more consistent relationship with withdrawal timing compared to other factors.

## 3 Results and Discussion

### 3.1 Climatology Outlook

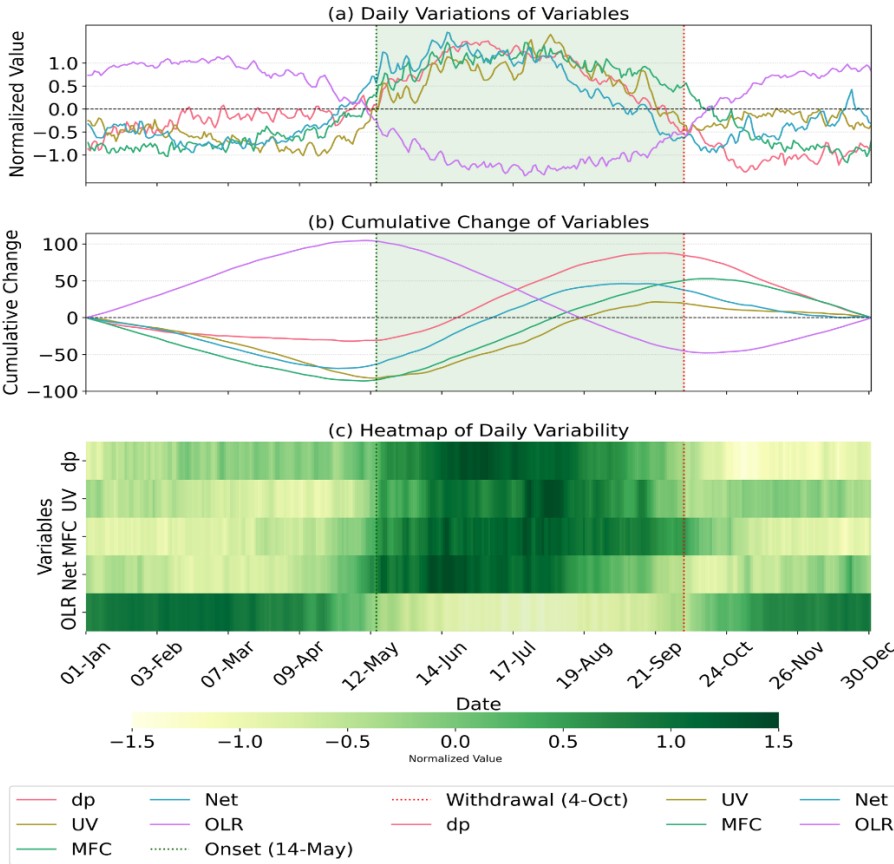

*Fig. 2 Daily and cumulative variations of monsoon parameters with their seasonal progression. (a) Daily variations of normalized parameters: pressure gradient (dP; red), wind shear (UV; yellow), moisture flux convergence (MFC; green), net precipitation (Net; blue), and outgoing longwave radiation (OLR; purple). (b) Cumulative changes of the same parameters with identical color coding. (c) Color strip timeseires showing the daily variability of all parameters throughout the year. Vertical dotted lines indicate monsoon onset (green; 14-May) and withdrawal (red; 4-Oct), with light green shading highlighting the monsoon active period in (Fig a nd b). All parameters are normalized and calculated according to Eqs. (1) and (2). This figure was created using Python 3.10 with Matplotlib 3.5.2 (https://matplotlib.org/) and Seaborn.*

Fig. 2 explained how the MSwM (CPM) index is constructed by combining both thermodynamic and dynamic climatology daily contribution (Fig. 2.a) and their cumulative change (Fig. 2.b) of same variables. Cumulative change curves (CMFC, Cdp, Cwind) help track the transitions in atmospheric conditions that define the onset and withdrawal of the monsoon. The simultaneous positive and negative shifts in MFC, OLR, pressure differentials, and wind shear facilitate the identification and calculation of monsoon onset and withdrawal. Both figures underscore the significance of cumulative effects in the MSwM index, where prolonged alterations over several days in moisture flux, wind shear, and pressure differentials signify critical transitions in the monsoon cycle, thereby illustrating the seasonal progression of the monsoon in contrast to mere daily variations. The Color strip timeseries (Fig. 2.c) support more clarity transaction of monsoon season by same variables values. The climatology dates for each year are shown in Table S-1 and S-2 of the supplemental material.

However, the small asynchrony among variables in Fig. 2 arises because each diagnostic reflects different aspects of the monsoon system with distinct adjustment timescales: dynamical fields (wind shear, pressure gradient) respond rapidly to convective heating through Gill-type circulation, while thermodynamic fields (MFC, TpNet, OLR) involve moisture storage and cloud–radiation feedbacks that introduce short lags (Gill, 1980; Wang et al., 2009). The CPM index minimizes this effect by averaging across all five variables, so that the central onset and withdrawal dates are robust, while the spread provides an objective measure of uncertainty.

Some studies have indicated that the monsoon onset over the Bay of Bengal is significantly correlated with that over the South China Sea and India (Xing et al., 2016). The India Monsoon Index (IMI) , the Webster and Yang monsoon index for Asia (WYI), the West North Pacific monsoon index (WNPMI), and are some of the well-known monsoon indicators for the South Asian region (Goswami et al., 1999; Wang et al., 2001, 2004; Webster and Yang, 1992). However, seasonal wind variation and uniform rainfall can also be used to designate MSwM zones as sub-regions (Oo, 2022a, 2023b). In terms of annual variability, MSwM and other South Asian monsoon indicators show a comparable time-series pattern and a positive moderate connection (Supplementary Fig S-6).

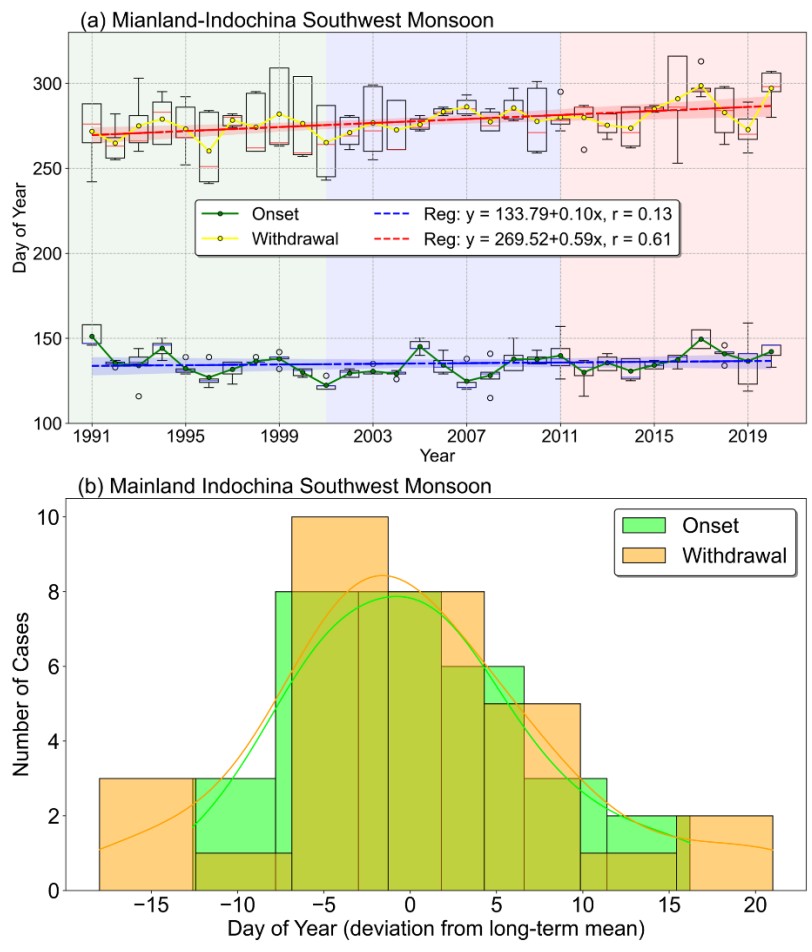

Fig. 3 (a) Interannual variability of MSwM onset (green line) and withdrawal (yellow line) dates, with trends. (b) Frequency distribution of deviations from mean onset and withdrawal dates, with implications for Indochina agriculture. This figure was created with Python 3.10 (Matplotlib 3.5.2 [https://matplotlib.org/]).

Examining the distribution patterns of the onset and withdrawal dates of the MSwM across MIC is interesting, despite the MSwM index reflecting changes in the whole MIC rather than a specific region within its domain. In this study, we only consider interannual variability over southern region (95E-100E, 10N-15N) ("S" area in Fig. 1.b) where is the first onset point (during onset) and last withdrawal point (during withdrawal) in north-south-north shifting of monsoon characteristic due to its role in the migration of the Intertropical Convergence Zone (ITCZ), which shifts northward during boreal summer, initiating intense convective activity and precipitation. At this latitude, the strong land-ocean thermal contrast generates a pressure gradient, drawing moist southwesterly winds from the Indian Ocean that converge and bring rainfall (Goswami and Xavier, 2005). This region aligns with the early onset of monsoon rainbands and moisture convergence observed in climatological data, as well as the geographical position of southern Myanmar, India and Sri Lanka, which are the first landmasses to experience the advancing monsoon (K Lau, 2000). Fig. 3.a shows interannual variation of onset and withdrawal dates with their trend including whisker statistical box. It indicates the timing of onset and withdrawal phases, which are vital for understanding how the regional monsoon system

is developing. Over the MIC, early or delayed onset and withdrawal of the monsoon can dramatically
affect the seasonal rainfall patterns, which may lead to regional crop production and society plans. The
trend lines in both phases suggest possible long-term shifts in monsoon behavior (Fig. 3.a), may be
influence of the broader climatic drivers such as variability of ENSO or Indian Ocean dipole (IOD)
(Ding et al., 2011a; Wang and Ho, 2002). While ENSO/IOD influence monsoon dynamic circulation,
their direct impact on MSwM onset timing is secondary to regional thermodynamics (Oo, 2021, 2022b;
Oo et al., 2025; Oo and Jonah, 2024). The time-series of dynamic and thermodynamic trend displayed
that withdrawal dates are significantly greater variation than onset dates within five variables of CPM
index for each year especially in dynamic boundary (Fig. 4). The frequency distribution of deviations
from the mean onset and withdrawal dates (Fig. 3.b), which explained that onset and withdrawal date
may early or delay generally one to two weeks (5 to 7 days as usual in general). The longest delay
(early) withdrawal phases occurred with 20 days (15 days) during this 30-year study period 1991-2020.
The onset phases are generally characterized by a rapid shift in moisture flux and dynamic
transformations over MIC, whereas the withdrawal phases occurs more gradually and may be affected
by extensive atmospheric patterns (Seager et al., 2010), including modifications in subtropical jets,
mid-latitude disturbances, and tropical easterly waves, which can introduce variability in the timing of
the retreat (Hu et al., 2019).

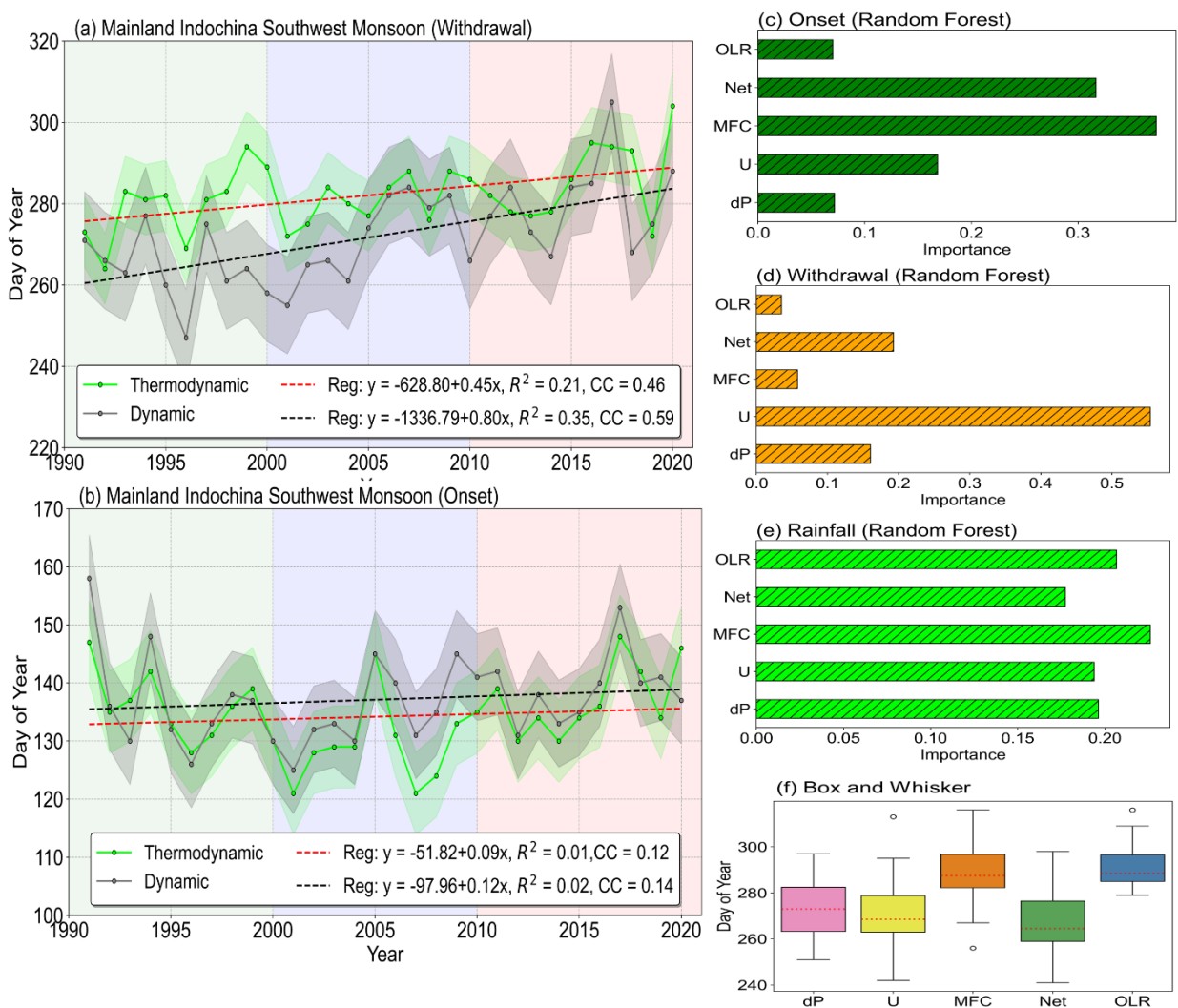

*Fig. 4 Interannual variation of thermodynamic and dynamic factors during (a) onset and (b) withdrawal phases of MSwM, with trends, highlighting the impact on mainland Indochina. And their sensitivity tests by random forest method (c) for CPM index onset, (d) for CPM index withdrawal and (e) for monsoon regional rainfall. (f) The boxes whisker plot of five physical parameters to determine the MSwM onset and withdrawal (Oo etl. 2023). This figure was created with Python 3.10 (Matplotlib 3.5.2 [https://matplotlib.org/]).*

The interannual variations of the Mainland Indochina Southwest Monsoon (MSwM) onset and withdrawal dates from 1991 to 2020 reveal a clear divergence in behavior between the two phases when analyzed through thermodynamic and dynamic components using the CPM index (Fig. 4). The onset phase shows minimal long-term change, with weak regression slopes of +0.09 days/year for the thermodynamic component and +0.12 days/year for the dynamic component, both statistically insignificant (Fig. 4.b). In contrast, the withdrawal phase exhibits a significant delay, especially in the dynamic processes, with a regression slope of +0.80 days/year and a moderate correlation ($R^2 = 0.35$, CC = 0.59). The thermodynamic component also shows a positive trend, albeit weaker, at +0.45 days/year ($R^2 = 0.21$, CC = 0.46), indicating that dynamic atmospheric factors, such as upper-level wind changes, are increasingly contributing to the delayed monsoon withdrawal (Fig. 4.a).

The random forest analysis further supports these findings. For onset prediction, the Net heat
flux (Net) and Moisture Flux Convergence (MFC) are the most important factors, reflecting the
dominant role of thermodynamic processes (Fig. 4.c). For withdrawal prediction, however, the 850-
284 hPa zonal wind (U) emerges as the most critical driver, followed by pressure gradient (dP), with MFC
playing a secondary role (Fig. 4.d). Regarding seasonal rainfall, all five parameters (U, dP, MFC, Net,
OLR) contribute relatively evenly (Fig. 4.e), highlighting the coupled influence of both dynamic and
thermodynamic factors on rainfall variability.

The box-whisker plot (Fig. 4.f) shows that MFC and OLR tend to correspond with delayed
withdrawal dates, suggesting that lingering moisture convergence and persistent convective activity
can postpone the withdrawal phase. This aligns with previous findings that regional convective
systems and late-season tropical cyclones (Akter and Tsuboki, 2014; Fosu and Wang, 2015; Oo et al.,
2024) can sustain rainfall events even after large-scale monsoon winds weaken (Chou et al., 2009).
These results collectively point to a dynamic-thermodynamic asymmetry: while monsoon onset is
controlled primarily by energy build-up and moisture availability, the withdrawal is increasingly
modulated by dynamic atmospheric circulation anomalies, such as upper-level wind changes and
tropical disturbances from the South China Sea (Wang and Zhou, 2024).

The analysis suggests that the two phases of monsoon are influenced by distinct mechanisms
with weak interdependence. The onset phase remains stable over the study period, primarily driven by
thermodynamic factors, while the withdrawal phase shows a significant delay due to dynamic factors
(Fig. 4). This decoupling might be explained by different large-scale climate processes governing the
two phases: onset is mainly linked to pre-monsoon land-sea thermal contrasts and moisture build-up,
whereas withdrawal is more sensitive to post-monsoon circulation shifts, tropical cyclone activity, and
upper-level wind anomalies. However, this finding also highlights the need for further research into
potential indirect links, such as how early or late onsets may influence intra-seasonal rainfall breaks,
which in turn could modulate withdrawal characteristics.

## 3.2    Variation of MSwM withdrawal dates and Rainfall in October

The first Empirical Orthogonal Function (EOF) modes of October rainfall and mean sea level
pressure over the study area, and their normalized principal components (PCs) expressed in Fig. 5. The
first EOF for rainfall, explaining 34.4% of the variance (Fig. 5.a) and the first EOF for MSLP,
explaining a larger 81.5% of variance, indicating its stronger influence on regional climate (Fig. 5.b).
Positive and negative eigenvectors suggest the impact of MSLP and rainfall distribution over
withdrawal phases that reduction in rainfall and increasing in pressure. The regression between
monsoon withdrawal dates by MSwM definition index (CPM) and regional rainfall explained positive

relations (green areas in Fig. 5.c) suggest that the index can significantly reflect the October rainfall

over the study area with 95% confidence. This show CPM index is significantly reflected to southern

MIC ("S" area in Fig. 1.b), where is the last point of monsoon withdrawal, regional rainfall during

withdrawal phases. Moreover, PCs time series of rainfall (RF), and SLP, from 1991 to 2020 (Fig. 5.d),

are comparing with monsoon withdrawal dates and the correlation between withdrawal dates and SLP

shows 0.41, and between RF exhibited 0.24, with statistically confidence ($p > 0.05$). However, the

weak correlation between withdrawal timing and PCs RF suggests that while the timing of monsoon

withdrawal affects the overall seasonal rainfall, it does not directly influence the spatial distribution of

rainfall. This is because spatial distribution is primarily governed by local factors such as topography,

moisture transport, and mesoscale atmospheric dynamics rather than the withdrawal timing alone. A

late withdrawal may extend the period of rainfall over certain regions, increasing total rainfall. This

dominant EOF modes capture the large-scale spatial variability of October rainfall and sea level

pressure pattern, which is vital for understanding the dynamics of the transition from the warm wet

southwest monsoon to the cold dry northeast monsoon season over MIC (Hannachi, 2004; Oo, 2022c;

Wu and Mao, 2018).

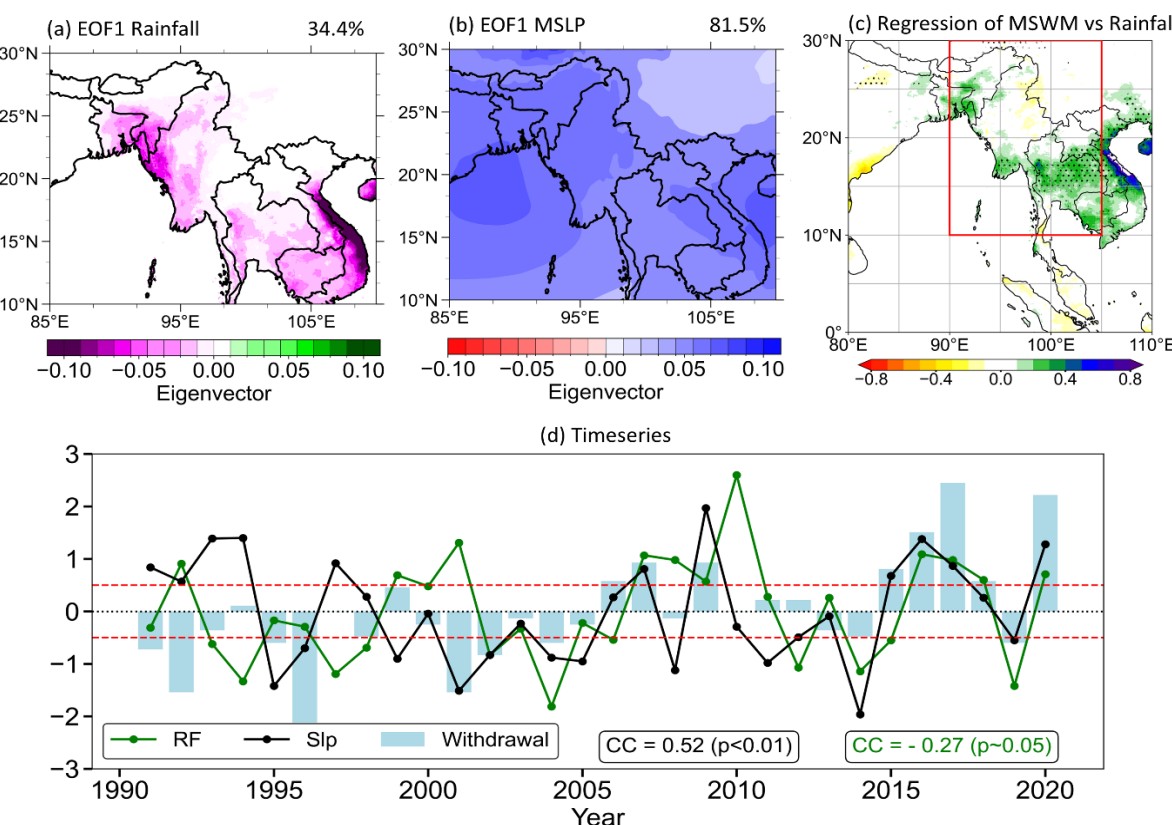

*Fig. 5 First EOF modes of (a) rainfall and (b) Slp. (c) The regression values of withdrawal CPM indexd and regional october rainfall with dotted area of 95% statistically confident by t-test. (d) The interannual varaition of normalized PCs of first two EOF and normalized MSwM withdrawal dates with their correlation CC by respective color. The horizontal red dotted sperated the late (>0.5) and early (< -0.5) witdrawal years by their normalized anoamlies varues. This figure was created with Python 3.10 (Matplotlib 3.5.2 [https://matplotlib.org/], Cartopy 0.20.0 [https://pypi.org/project/Cartopy/]).*

In addition, the SLP patterns are directly related to the atmospheric circulation that initiatives rainfall and weather conditions over the region (Loikith et al., 2019). The shift in the SLP pattern could indicate changes in the positioning of the low-level monsoon winds and subtropical high-pressure systems, which bring the moisture-flux into mainland Indochina (Liu et al., 2021) (Fig S-5 in supplementary ). The PCs associated with these modes provide a temporal perspective, indicating how these dominant patterns advance over time. To perform composite analysis we collected eight delay withdrawal years (2006, 2007, 2009, 2015, 2016, 2017, 2018 and 2020) and eight early withdrawal years (1991, 1992, 1995, 1996, 2001, 2002, 2004 and 2019) by anomalies timeseries with PCs, we collected positive(negative) 0.5 ( +/- renormalize 5-7 days) values years into late (early) withdrawal years.

## 3.3 Composite

The composite anomalies analysis of three majors' variables what are used to define monsoon onset and withdrawal are explained in Fig. 6. The climatological values between early years (first column) and late years (middle column), with their percentage difference (last column) over mainland Indochina, were compared. The analysis indicates notable patterns in the distribution of monsoonal rainfall, especially in southern MIC. The difference % map delineates areas where rainfall has either diminished or increased, namely over southern MIC (Fig. 6.a and b). Their different percentages also result significantly in the same region as shown in Fig. 6.c. This confirmed that the most accurate classification skill of the MSwM CPM index over this southern MIC region as in (Fig. 5.c).

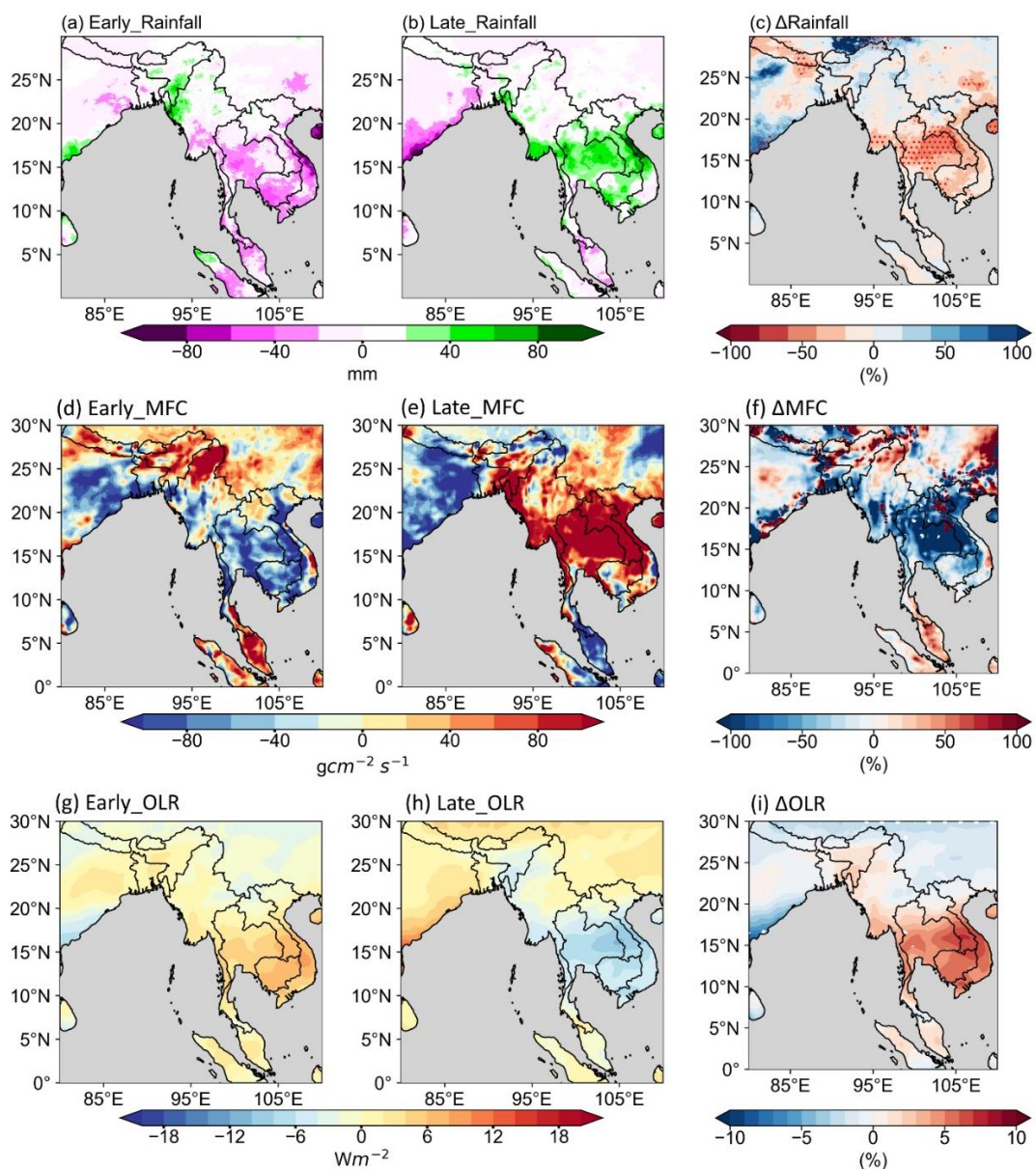

*Fig. 6 Climatological anomalies mean rainfall (mm), MFC (g/cm²/s) and mean OLR (Wm⁻²) for (a,d,g) early years, (b,e,h) late years, and (c,f,i) the percentage difference, illustrating changing moisture dynamics over mainland Indochina. Red dotted show the area of 95% statistically confident by t-test. This figure was created with Python 3.10 (Matplotlib 3.5.2 [https://matplotlib.org/], Cartopy 0.20.0 [https://pypi.org/project/Cartopy/]).*

Changes in Moisture Flux Convergence (MFC) also impact rainfall patterns, with decreased MFC potentially reduction rainfall and increasing it, leading to wet conditions (Fig. 6.c and d). The figure compares climatologically to mean MFC in low-lying areas over southern MIC show similar negative/ positive patterns is validated by their different values (Fig. 6.f). Same patterns are also found for OLR of early and late withdrawal years over southern MIC. Thus, the MSwM CPM index is significantly reflected in this area.

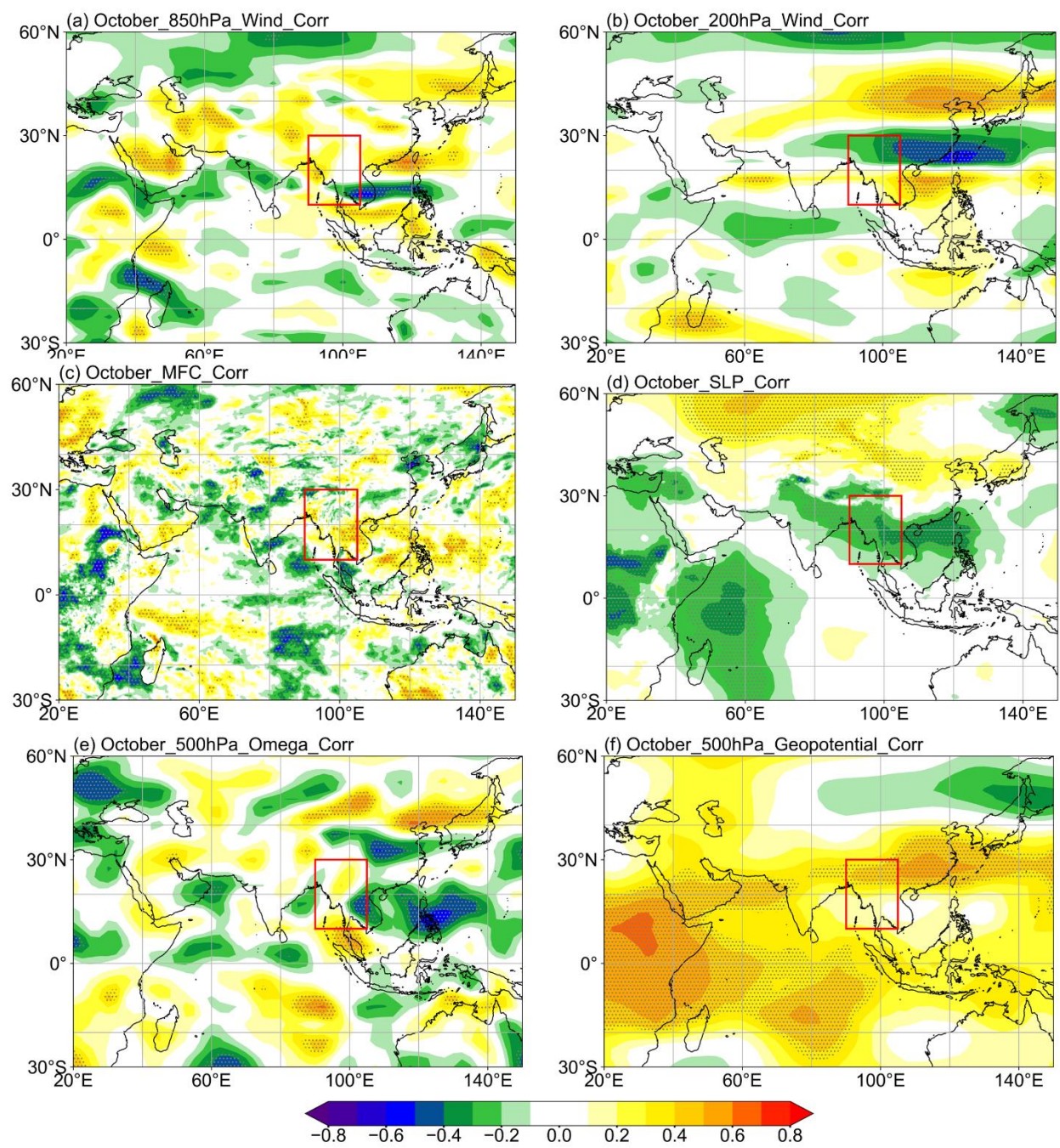

Fig. 7 Correlation between withdrawal date and October (a) 850 hPa wind, (b) 200 hPa wind, (c) MFC, (d) SLP, (e) 500 hPa Omega, and (f) 500 hPa geopotential, highlighting the drivers of monsoon withdrawal in mainland Indochina. Dotted shows the area of 90% statistically confident by t-test. This figure was created with Python 3.10 (Matplotlib 3.5.2 [https://matplotlib.org/], Cartopy 0.20.0 [https://pypi.org/project/Cartopy/]).

The correlation between these various atmospheric variables, showing how relation of these variables with the delayed monsoon withdrawal (Fig. 7). The correlation between the withdrawal date and wind speed was calculated at each grid point, analyzing the withdrawal date against both the u-component and v-component speed. Statistical significance was determined using a student's t-test, with the dotted areas marking regions where the correlation is significant at the 90% confidence level. The correlations for 850 hPa wind speed (Fig. 7.a) expose strong negative relationships over MIC, indicating that weaker low-level winds contribute to the delay in withdrawal. This aligns with the

positive-negative-positive trend pattern as in the Fig. 8.a and c, where negative correlations suggest a weakened low-level wind over MIC. In contrast, the 200 hPa wind correlation (Fig. 7.b) shows a positive relationship, particularly over the northern regions, suggesting stronger upper-level winds during delayed monsoon retreat periods, which likely strengthens the subtropical westerly jet (SWJ) region and weakening in Tropical Easterly Jet region (TEJ). The SWJ, defined as a dominant westerly wind stream at approximately 200 hPa in mid-latitudes, and the TEJ, a tropical Easterly wind at similar altitudes. Similar patterns are also exhibited in trend plots Fig. 8. b and d. A delayed withdrawal sustains the thermal gradient between the Indian Ocean and the Asian continent, maintaining a strong meridional temperature gradient in the upper troposphere and thereby intensifying the SWJ. Simultaneously, the TEJ weakens due to reduced upper-tropospheric divergence and the diminishing impact of tropical heating as the monsoon season transitions.

The correlation with Moisture Flux Convergence (MFC) (Fig. 7.c) also specifies a significant positive relationship in key study areas and positive trends also exhibited over same area (Fig S-7 in supplementary). This positive trend suggests that delayed monsoon withdrawal is associated with stronger moisture convergence, trapping moisture likely to experience rainfall in southern MIC for a longer period and it's also association with previous composite analysis as in Fig. 6. The Sea Level Pressure (SLP) correlation (Fig. 7.d) also shows a study area of negative correlation, which suggests that lower pressure systems dominate during delayed withdrawal, promoting cyclonic activities that extend the monsoon season and rainfall. Meanwhile, the positive correlations with 500 hPa Omega (Fig. 7.e) highlight the role of vertical motion over southern MIC, where positive Omega values (upward motion) correlate with a delayed withdrawal, can lead to cloud formation and rainfall if the conditions are right. Moreover, the 500 hPa geopotential positive correlations (Fig. 7.f) also show a weakened mid-tropospheric ridge over the subtropics with positive trend (Fig S-7 in supplementary), leading to the late monsoon withdrawal as the atmospheric circulation shifts.

The wind trends and anomalies highlight a significant alteration in both the lower (850 hPa) and upper (200 hPa) wind patterns (Fig. 8). The 850 hPa wind pattern (Fig. 8.a) indicates a weakening easterly flow over the South China Sea and southern MIC, and the 200 hPa wind trend (Fig. 8.b) indicates an intensification of the westerly flow linked to the SWJ, enhancing the upward motion and which may lead to anomaly lower-upper dynamic circulation patterns, and it may lead to delaying the timing of seasonal withdrawal of the monsoon. There is a noticeable positive-negative zonal wind anomaly pattern, especially at the 200 hPa level, in the difference in wind structure between late and early years (late years minus early years) at both altitudes, and this pattern changes significantly over time (Fig. 8.c and d). Delays in the MSwM withdrawals are directly affected by changes in jet stream dynamics, such as the strengthening of the SWJ and the weakening of the Tropical Easterly Jet (TEJ).

The results of these additional investigations provided confirmation of this pattern of dynamic abnormality. Specifically, across the SWJ and TEJ regions, variations in wind intensity and direction are critical in affecting the delayed withdrawal trend, according to the CPM index analysis of these dynamic circulation patterns. Important regions where wind anomalies are strongly linked to delayed withdrawal are highlighted by the plus and minus signs in the Fig. 8. This emphasizes as they indicate critical areas where wind anomalies are closely associated with delayed withdrawal.

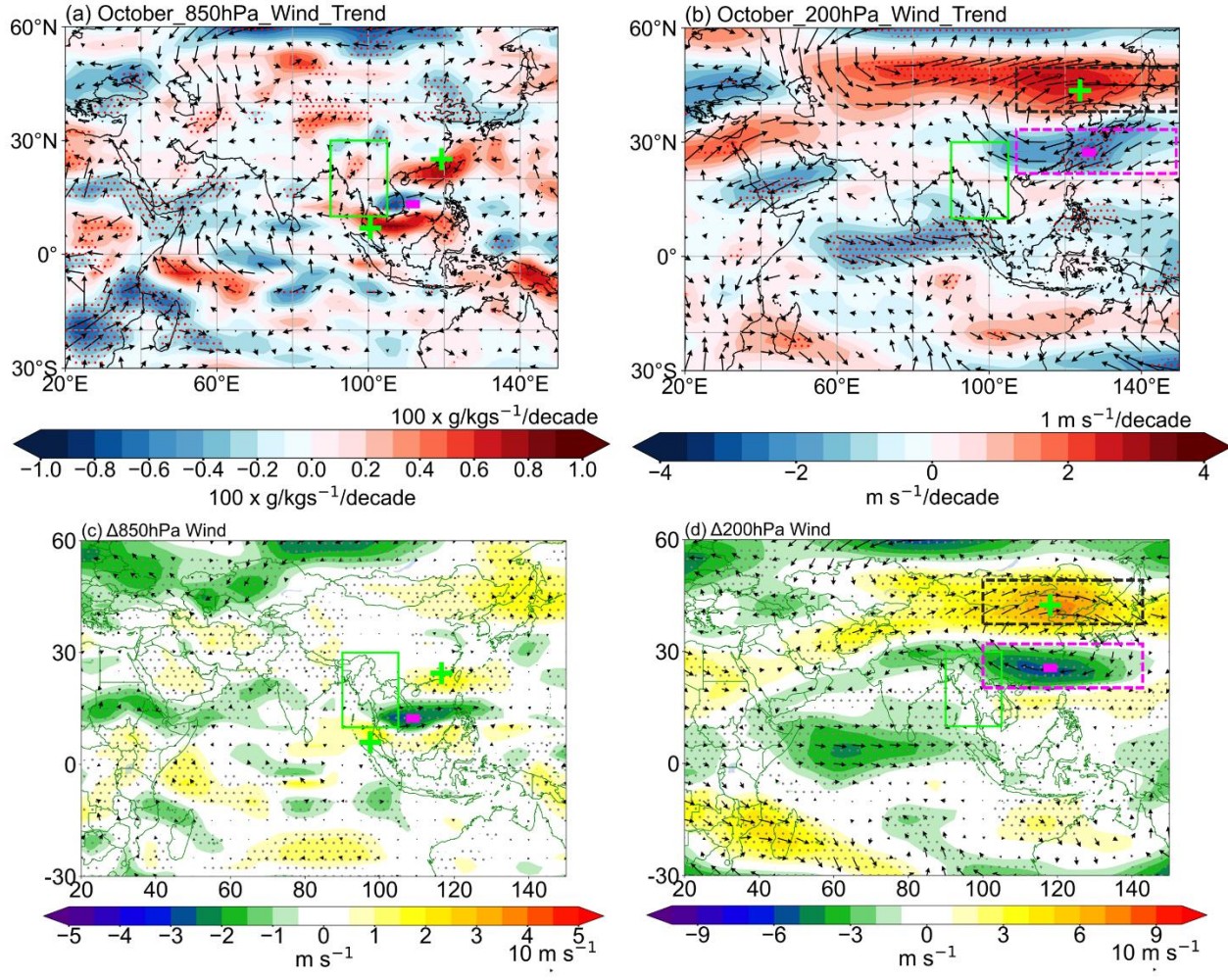

*Fig. 8 The spatial trend of (a) 850hPa horizontal moisture transport(g/kgs⁻¹) and 200hPa wind component (m/s). Percentage difference (late minus early) in horizontal wind patterns at (c) 850 hPa, and (d) 200 hPa between late and early years of MSwM withdrawal month October during 1991-2020. Red and grey dotted show the area of 95% statistically confident by t-test. This figure was created with Python 3.10 (Matplotlib 3.5.2 [https://matplotlib.org/], Cartopy 0.20.0 [https://pypi.org/project/Cartopy/]).*

The vertical structure of zonal wind, vertical motion, and moisture transport, comparing early and late years of the monsoon are exhibited in Fig. 9. The cross-section of vertical velocity over mainland Indochina, which is essential for understanding how wind circulation at different atmospheric layers contributes to vertical motion and convective processes (Kotal et al., 2014; Sawyer, 1947). The weak upward motion over MIC had occurred during the early years (Fig. 9.a) and found exceeds and shifts northward during the late years as a reverse (Fig. 9.b). This reflects a strengthening

in monsoon intensity, and this is consistent with the observed weakening of TEJ, which decreases
upper-level divergence and leads to delayed monsoon withdrawal.

The strong walker circulation over the study regions in the early years (Fig. 9.c), and weakens
in the late years (Fig. 9.d) are suggesting that the significant of TEJ and vertical circulation have
declined, contributing to the delayed monsoon withdrawal. The reduced convective activity and
moisture transport highlights how weaker jets are affecting monsoon dynamics (Roxy et al., 2015).

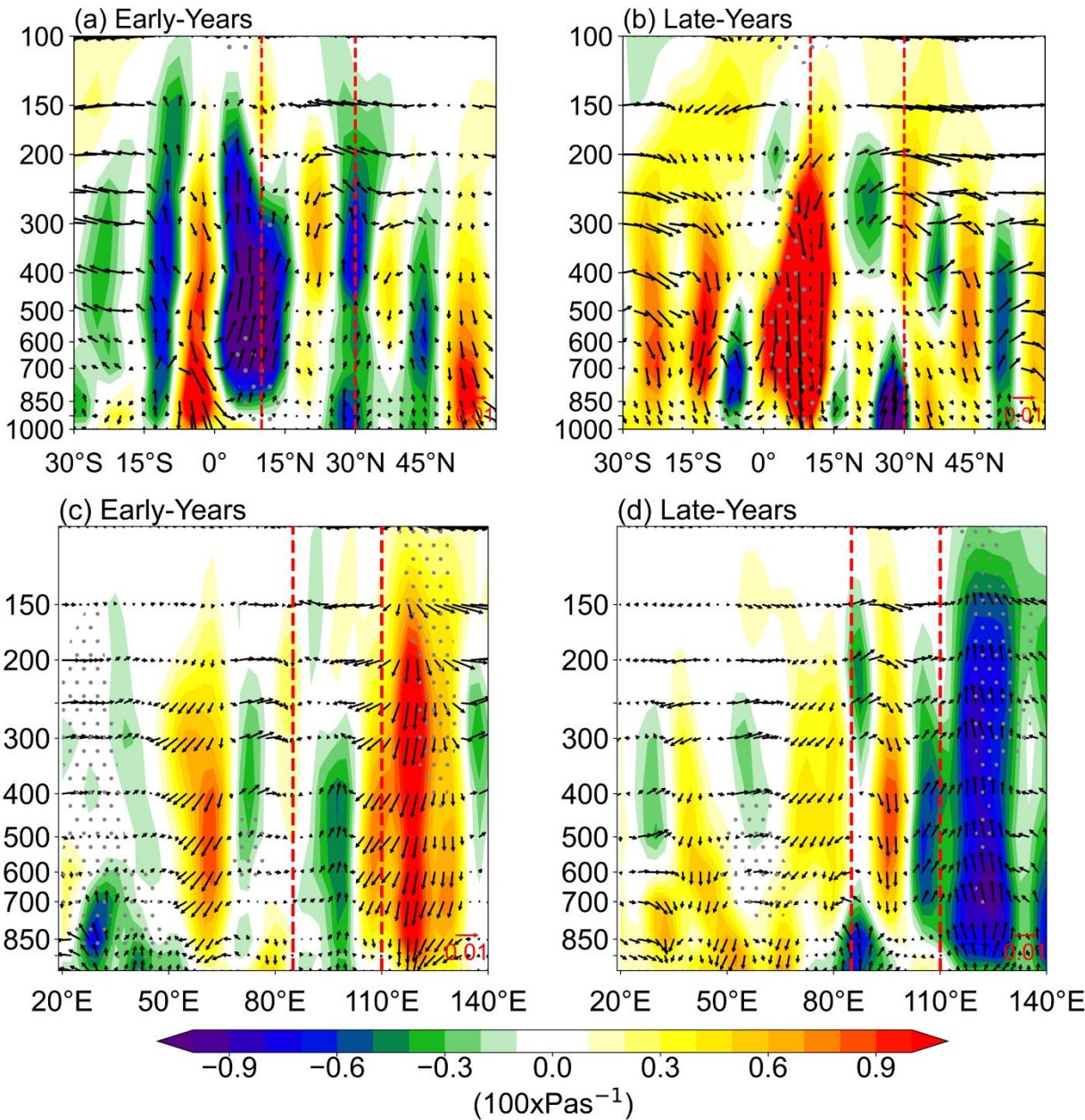

*Fig. 9 Vertical cross section of omega (Pas⁻¹) of longitudinal averaging over (90°E-110°E) and latitudinal averaging (10°N-30°N)*
*(a,c) for early years , (b,d) for late years. Grey dotted show the area of 95% statistically confident by t-test. This figure was created*
*with Python 3.10 (Matplotlib 3.5.2 [https://matplotlib.org/], Cartopy 0.20.0 [https://pypi.org/project/Cartopy/]).*

438   The weakening of the Tropical Easterly Jet (TEJ) and the concurrent intensification of the
439 Subtropical Westerly Jet (SWJ) exert a pivotal control on the monsoon withdrawal process through

modifications of the upper-tropospheric thermodynamic and dynamic structures. A pronounced negative trend in the 200 hPa zonal wind over the tropical belt signifies a weakening TEJ, while an enhanced westerly anomaly over the subtropics indicates a strengthening SWJ (Fig. 8.b and d). This shift reflects a northward migration of the jet core and a weakening of the upper-level Easterly ventilation, which reduces the divergent outflow critical for maintaining deep convection during the mature monsoon phase. The low-level wind trends (Fig. 8.a and c) depict a weakening of the 850 hPa monsoon westerlies, leading to reduced moisture convergence over the Indo-China Peninsula, as supported by the negative moisture flux convergence correlations. Furthermore, the suppressed ascending motion at mid-troposphere levels (Fig. 7.e), coupled with positive 500 hPa geopotential height anomalies (Fig. 7.f), signify the onset of mid-level atmospheric stabilization and the collapse of the monsoon thermal structure.

The vertical cross-sections reveal that during the late years, corresponding to delayed withdrawal events, the upper-tropospheric divergence weakens (associated with TEJ weakening), while the upper-level westerly shearing and subsidence induced by the intensified SWJ strengthen (Fig. 9.c and d) This enhanced subsidence promotes tropospheric drying and suppression of convection, which together act as a dynamical brake on the monsoon system, facilitating its withdrawal. Collectively, these findings exhibited the barotropic and baroclinic adjustments in the upper-level circulation, where the interaction between the weakening TEJ and the intensifying SWJ modifies the large-scale monsoon dynamics, disrupts the monsoon Hadley circulation, and accelerates the seasonal transition toward the dry post-monsoon regime. This conclusion lends credence to those earlier findings (Krishnamurti et al., 2012; Roxy et al., 2015). In addition, prior research has demonstrated the connection between sustained moisture transport and extended convective activity with the monsoon, which is supported by the positive link between moisture flux convergence and delayed monsoon withdrawal (Goswami et al., 2006). The atmospheric dynamics anomaly, specifically the weakening of the TEJ and the intensification of the SWJ, are significant variables influencing the noted trend of delayed monsoon withdrawal.

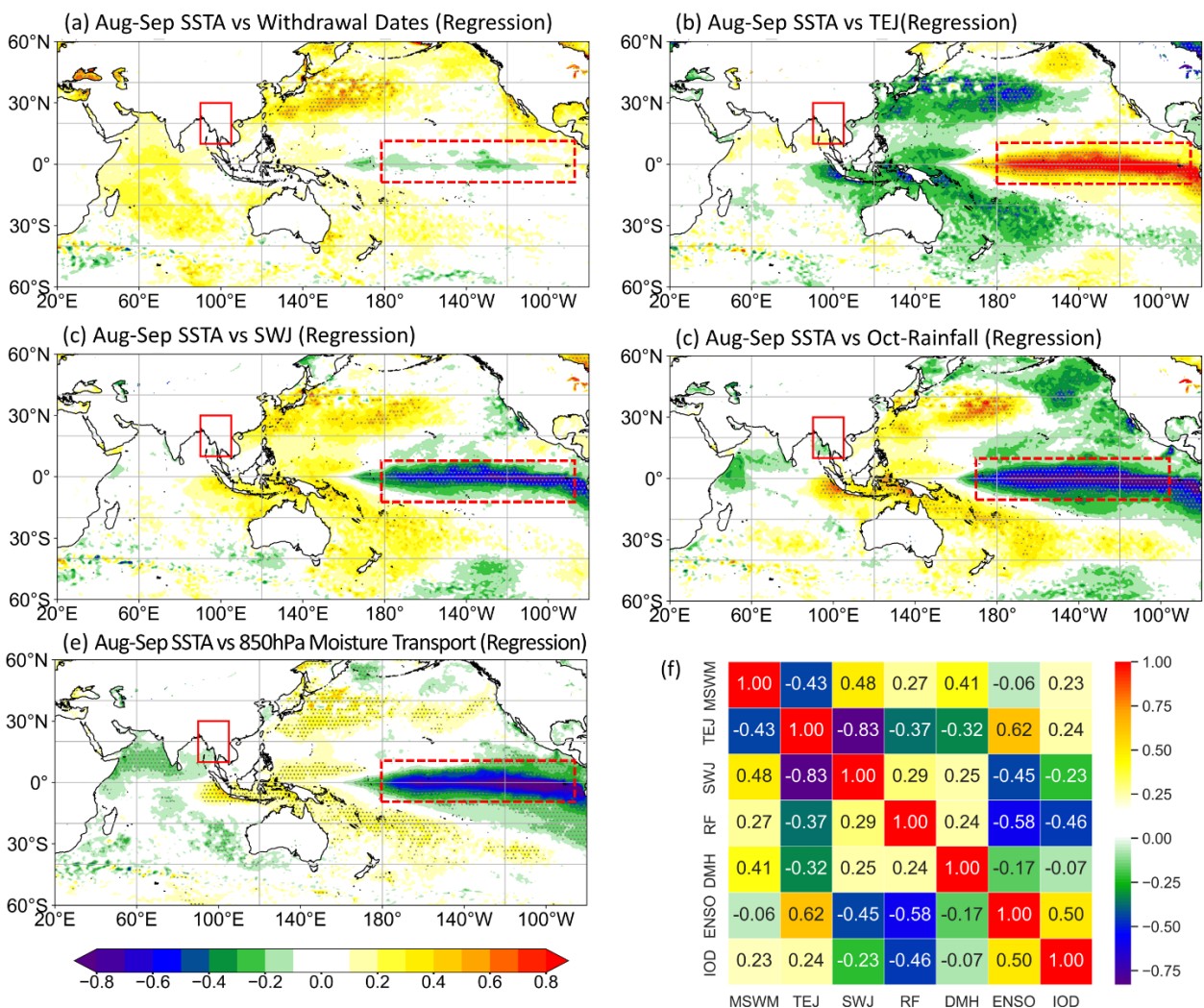

*Fig. 10 Regression between Aug-Sep Sea surface temperatures (SSTs) and (a) MSwM withdrawal dates, (b) October tropical easterly jet, (c) October sub-tropical westerly jet, (d) October rainfall over MIC and (e) October 850hPa moisture divergent. Dotted hatches mean 95% confident area by t-test statically. The red boxed show MSwM region and red dotted box show the area in the Pacific with the strongest negative positive correlation. (f) Correlation heatmap between variables used in this study. DMH refers to the MSwM withdrawal dates from National weather services recorded. This figure was created with Python 3.10 (Matplotlib 3.5.2 [https://matplotlib.org/], Cartopy 0.20.0 [https://pypi.org/project/Cartopy/]).*

The relationship between August-September SST anomalies and the delayed withdrawal of the MSwM showing not significant correlation over the equatorial Pacific Ocean (Fig S-9, supplementary), indicating that negative anomaly SSTs in this region are associated with delayed monsoon withdrawal (Fig. 10.a). This is constant with the role of warm SSTs over Indochina region are maintaining convective activity (Roxy et al., 2015; Krishnan et al., 2016) and preventing the on-time withdrawal of the monsoon however cold SSTs over Niño3-4 region does not directly impact on withdrawal dates. The red dotted boxed region shows the area in the Pacific with the strongest negative/positive correlation, suggesting a link between SST anomalies in the central Pacific and the timing of monsoon withdrawal. The relationship between SST and the tropical easterly jet (TEJ) and subtropical westerly jet (SWJ) in October, the strong positive correlation between SST and TEJ over the Pacific Ocean (Fig S-9.b, supplementary) suggests that warmer SSTs exceeding the strength of TEJ (Fig. 10.b). This

agreed with the previous trend and composite finding (Fig. 8) that the weakening of the TEJ is a critical factor in delaying the monsoon withdrawal. The weakening of the TEJ boost the lower-level monsoon circulation to endure for an extended duration over MIC (Huang et al., 2020; Sreekala et al., 2014). In contrast, Fig. 10.c shows a negative regression between SST and SWJ, this demonstrates that cooler SSTs over same area also strengthen the SWJ. This finding supports the idea that a positive anomalies SWJ also impact to delayed withdrawal (Dimri et al., 2015; Sreekala et al., 2014).

The Aug-Sep SST of tropical Pacific and Indian ocean and rainfall within October can also predict to MIC October rainfall. The negative correlation otherwise (La Niña) in the equatorial Pacific and the negative Indian Ocean Dipole (IOD) mode are associated with exceeding rainfall over MIC (Fig. 10.d), which is a mark of a extended monsoon (as mentioned in Fig. 4). This association supports the earlier finding that increased SSTs are associated with extended rainfall during the late monsoon, especially in the central Pacific and the Indo-Pacific Warm Pool (Ghosh et al., 2009; Sabeerali et al., 2014). Furthermore, the pattern of connection associates with the impact of global climate models like the El Niño-Southern Oscillation (ENSO), which changes regional SSTs and rainfall distributions in the Indo-Pacific area.

To confirm this SST anomaly influence over regional rainfall or moisture flux patterns, we performed the correlation between 850-hPa moisture transport strength over MIC and Indo-Pacific SST (Fig S-9.e, supplementary). The negative regression coefficients over the central Pacific and the northern western Indian Ocean indicate that negative ENSO and IOD enhance moisture transport at lower levels (Fig. 10.e). However, ENSO significantly influences the monsoon onset in the Indochina region, where El Niño tends to delay onset, as seen in the central Pacific's warm SSTs positively correlated and regressed with later onset (Fig.S-10). These vice versa correlation and regression results all together point to the critical role of SSTs in driving the extended moisture convergence that maintains convective activity and delays monsoon withdrawal (Roxy et al., 2019; Sharmila et al., 2013). While this study identifies ENSO and IOD as key modulators of MSWM onset and withdrawal, emerging evidence suggests that Arctic-monsoon teleconnections may also play a role. Recent work demonstrates that MSWM intensity anomalies can drive September Arctic Sea ice variability via atmospheric bridges (Than Oo et al., 2025). Moreover, Chen et al., (2024) and Cheng et al., (2025) highlight Arctic sea ice potential feedback on tropical modes (ENSO/IOD), which in turn affect monsoon dynamics. Although our analysis focuses on tropical drivers, the bidirectional nature of these interactions, particularly the Arctic's indirect influence on withdrawal via ENSO/IOD, and this should be prioritized for further investigation.

In addition, the correlation matrix in Fig. 10.f summarizes the links among the main variables of
the research, including the MSwM withdrawal index, TEJ, SWJ, rainfall (RF), 850-hPa moisture
transport, and indices indicative of ENSO and IOD. This exhibited the anomalous SSTs, especially in
the central Pacific and northern Indian Ocean, significantly influence the intensity of the TEJ and SWJ,
as well as moisture transport and rainfall patterns. The weakened TEJ, strengthened SWJ, and positive
moisture convergence led to the well-known delay of MSwM departure (Fig. 11). The results align
with the current literature connecting SST anomalies, major climate models like ENSO and IOD, and
monsoon variability (Ding et al., 2011b; Jia et al., 2013; Krishnamurthy and Kirtman, 2009).
Comprehending these linkages enhances long-term predictions and prepares agricultural systems for
modifications in the southwest monsoon departure date from MIC.

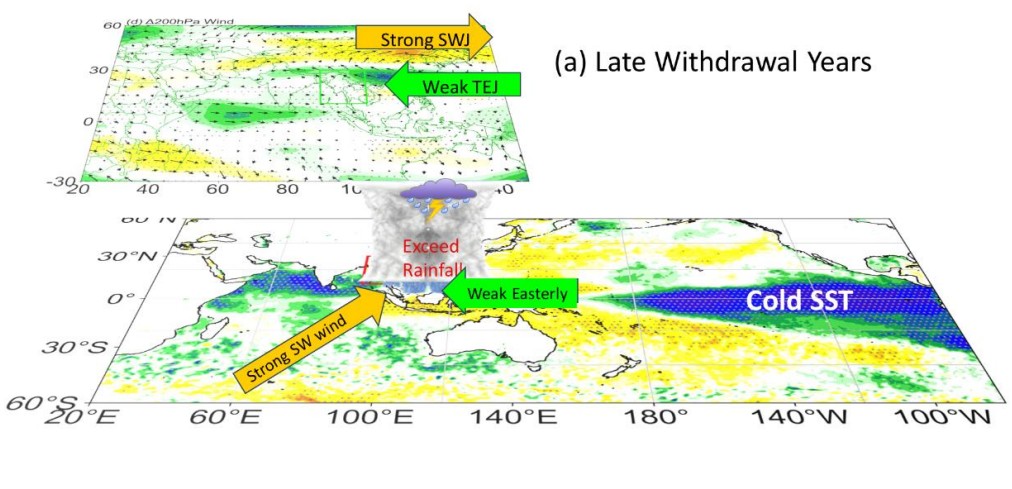

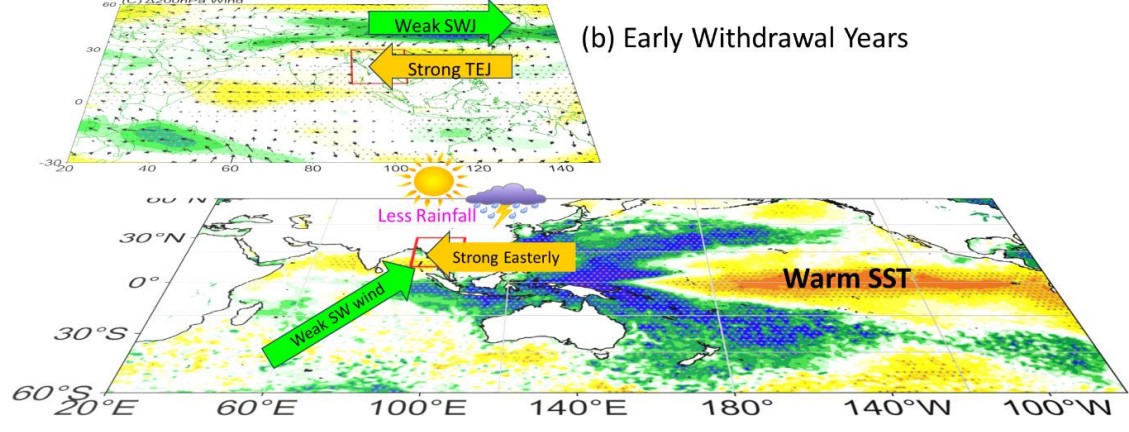

*Fig. 11 Air-Sea interaction Dynamical schematic of (a) late and (b) early withdrawal years. This figure was created with Python 3.10*
*(Matplotlib 3.5.2 [https://matplotlib.org/], Cartopy 0.20.0 [https://pypi.org/project/Cartopy/]).*

# 4   Conclusion

Focusing on the timing of the monsoon onset and withdrawal, the study offers vital insights into
the changing dynamics and interannual variability of the Mainland Indochina Southwest Monsoon

(MSwM). With the development of the Cumulative Change of the MSwM (CPM) index, a more
thorough knowledge of monsoon transitions may be achieved than with typical daily measurements.
This index effectively captures the continuous build-up of crucial atmospheric components.

Withdrawal timing has been noticeably delayed over the past few decades, according to the
findings, which also show clear patterns in the start and withdrawal phases. SWJ and the TEJ, which
control the monsoon withdrawal processes, have had a significant impact on this delay. Additionally,
the MSwM atmospheric circulation and moisture transport are significantly influenced by SST
anomalies, especially in the western Pacific and Indian Oceans. In mainland Indochina, extended
monsoon seasons increase the risk of flooding and interfere with agricultural cycles, underscoring the
urgent need for efficient water management and flexible farming techniques.

As conclusion, the MSwM CPM index is a great tool for tracking monsoon variability, and the
framework it gives for studying how climate change is affecting the regional monsoon system through
composite correlation and trend analysis is invaluable. Improving our understanding of monsoon
behavior and constructing more accurate prediction models will require further studies, specifically on
the teleconnection mechanisms between large-scale climatic drivers (such ENSO and IOD) and
MSwM.

## Data Availability

### Source Data

All Reanalysis rainfall, wind components, OLR, and Mean Seal Level Pressure netcdf4 data for this study were downloaded from the NCEP and ECMWF data portal.

The historical record of onset and withdrawal dates by DMH of Myanmar the actual monthly rainfall observation data and mean sea level pressure data from 79 observation stations used to support the findings of this study was provided under permission by Myanmar's Department of Meteorology and Hydrology (DMH) and hence cannot be freely distributed. Requests for access to these data should be made to the Director-General of DMH, Myanmar. https://www.moezala.gov.mm/

### Software availability

Open Grads (http://opengrads.org/ ), Climate data operator (https://code.mpimet.mpg.de/ ), Python and IBM SPSS are mainly used for this study. Among these first two are open-source applications for everyone. Codes are also available upon request.

## Conflicts of Interest

I declared that there is no potential conflict of interest with any of the following statements.

1.  For any component of the submitted work, the author received no cash or services from a third party (government, commercial, private foundation, etc). (including but not limited to grants, data monitoring board, study design, manuscript preparation, statistical analysis, etc.).

2.  The author is not affiliated with any entity that has a direct or indirect financial interest in the manuscript's subject matter.

3.  The author was involved in the following aspects of the project: (a) idea and design, or data analysis and interpretation; (b) authoring the article or critically reviewing it for essential intellectual content; and (c) approval of the final version.

4.  This work has not been submitted to, and is not currently being reviewed by, any other journal or publishing venue.

5.  The author has no patents that are broadly relevant to the work, whether proposed, pending, or issued.

6.  The author received no payment or services from a third party for any aspect of the submitted work (government, commercial, private foundation, etc). (including but not limited to grants, data monitoring board, study design, manuscript preparation, statistical analysis, etc.).

## Funding Statement

This study is supported by the National Natural Science Foundation of China (Grant 42088101).

## Acknowledgment

The author gratefully acknowledges the financial support from the National Natural Science Foundation of China (Grant 42088101). The researcher expresses special thanks to all Professors who approve and support this research and Nanjing University of Information Science for support to come out of this research. I would also like to extend my gratitude to Professor Haishan Chen from Nanjing University of Information Science and Technology, for supervising this paper and his other support during this research. The author acknowledges heartfelt thanks to the scientists of the ECMWF for supporting ERA5 datasets and the Department of Meteorology and Hydrology for providing the data of Myanmar. Additionally, the author would like to thank three reviewers for their constructive and insightful reviews and comments, which have significantly helped to improve the manuscript. First author Kyaw Than Oo would like to show his gratitude to Mrs. Moh Moh Zaw Thin from UIBE, China, and Mr. Phyo Sitt Thyn Kyaw for their physical and mental support for this work.

## Author Contribution

**Kyaw**: Conceptualization, methodology, data curation, writing- original draft preparation., visualization and investigation.

**Chen**: Supervision.

**Jonah**: Writing – review & editing.

**Du**: Writing – review & editing.

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
