# Peer review of "Physical drivers and trends of the recent delayed withdrawal of the Southwest Monsoon over Mainland Indochina"

_EGUsphere, 2025_

## Author Response (AR1)

**Authors Respond**

We sincerely appreciate the time and effort you have dedicated to reviewing our manuscript. Your insightful comments and constructive suggestions have significantly improved the quality of this work. We have carefully addressed each of your concerns in the revised manuscript, with detailed responses provided point-by-point above.

**Response to Reviewer 1 Comments**

We are particularly grateful for your suggestions on clarifying physical mechanisms, expanding discussions on Arctic connections, and improving the overall readability of the paper. Your expertise has been invaluable in shaping this study into a more robust and impactful contribution to the field.

Thank you again for your thoughtful review. We look forward to your feedback on the revised version.

**Q1 (Line 52):** *"Base on"* → *"Based on"*
**A1:** Thank you for catching this typo. We have corrected it to **"Based on"** in the revised manuscript. Line 52.

**Q2 (Line 55):** *Delete the parenthesis around "ISM" "the ISM"*
**A2:** We have removed the unnecessary parentheses as suggested. Line 52-57

**Q3 (Line 55):** *"ISM should be presented in its full name"*
**A3:** We have now defined **"Indian Summer Monsoon (ISM)"** at its first mention for clarity. Line 52-57

**Q4 (Lines 79-83):** "In addition to ENSO, recent studies have demonstrated that mid-high latitude systems also have significant impacts on ENSO, East Asian monsoon onset and withdrawal, which should also be briefly reviewed (Hu et al. 2020, 2025)."
**A4:** We appreciate this suggestion. A brief discussion of influences (citing Hu et al. 2020, 2025) has been added to the introduction. Line 82-85

**Q5 (Lines 226-227):** *"To examine the possible role of ENSO and IOD in modulating monsoon onset, the authors should present SST anomalies pattern regressed upon the monsoon onset index."*

**A5:** We have now included the regression analysis of SST anomalies against the monsoon onset index in the revised manuscript. Fig-S10 and Line 503-507

**Q6:** *"Whether there exists a connection between the MSwM onset and withdrawal? I suggest examining this issue and adding some discussions."*

**A6:** This is a valuable point. We have analyzed the relationship and added a new figure and sub paragraph discussing potential linkages between onset and withdrawal. Line 267-306

**Q7 (Line 390):** *"are suggests that" → "are suggesting that"*

**A7:** We have corrected this grammatical error to "are suggesting that." Line 432

**Q8:** *"The physical process for the impact of the weakening of the TEJ and intensification of the SWJ on the monsoon withdrawal could be examined in more details."*

**A8:** We have expanded the discussion on dynamical mechanisms to clarify how TEJ weakening and SWJ intensification modulate monsoon withdrawal. Line 439-460

**Q9:** *"Do Arctic sea ice anomalies exert impacts on monsoon withdrawal? Recent studies indicated that Arctic climate systems could influence IOD and ENSO (Cheng et al. 2025; Chen et al. 2024). I suggest adding some discussions for future study."*

**A9:** We agree this is an important direction. A paragraph on potential Arctic linkages (citing Cheng et al. 2025 and Chen et al. 2024) has been added to the "Future Work" section. Line 509-516

**Response to Reviewer 2 Comments**

*This manuscript investigated the factors modulating the withdrawal of the Southwest Monsoon over Mainland Indochina, as well as the recent interdecadal delay withdrawal. The topic is important, and the results are interesting. However, some issues remain to be explained before this manuscript can be published.*

*We sincerely thank Referee for the constructive and insightful comments, which have greatly improved the quality of our manuscript. Below, we provide detailed responses to each point and indicate how we have revised the manuscript accordingly.*

1. *Page 3 Lines 70-74: In addition to precipitation and circulation, the thermal and moisture characteristics of the atmosphere also serve as an important indicator for describing the progression of the monsoon season.*

   *Zhang H, Liang P, Moise A, et al. Diagnosing potential changes in Asian summer monsoon onset and duration in IPCC AR4 model simulations using moisture and wind indices. Climate dynamics, 2012, 39(9): 2465-2486.*

   *Song L, Hu P, Chen W, et al. Increasing Trend of Summer Monsoonal Rainfall Tied to the Extension of the South China Sea Summer Monsoon Duration. Atmospheric Science Letters, 2025, 26(7): e1308.*

   *A1: We appreciate the reviewer's suggestion to include thermal and moisture characteristics as indicators for monsoon progression. Following the advice, we have revised the text and incorporated the suggested references (Zhang et al. 2012; Song et al. 2025) in Lines 75–77.*

2. *Equation 2: This study takes multiple factors into account to characterize the seasonal evolution of the monsoon. However, this approach may render the physical meaning of the index somewhat ambiguous. How do these factors compare in terms of relative importance? Which factor plays the dominant role?*

   *A2: We thank the reviewer for raising this important point. To address it, we conducted an additional analysis using a machine learning approach to compare the relative importance of the factors. A new figure and a dedicated subsection have been added, and relevant discussions are now included in Lines 262–286.*

3. *Following up on the previous question, these factors were obtained by averaging over different spatial domains. What is the rationale for choosing these domains? In the article, the ranges are directly specified, but no explanation is provided for the selection. It should be noted that tropical atmospheric dynamics exhibit well-defined spatial and location relationship (Gill 1980). For instance, convective activity can excite westward-propagating equatorial Rossby wave responses, which in turn lead to the strongest southwesterlies being located to the west of the convection center (e.g., Wang et al. 2009).*

   *Gill A E. Some simple solutions for heat-induced tropical circulation. Quarterly Journal of the Royal Meteorological Society, 1980, 106(449): 447-462.*

   *Wang B, Huang F, Wu Z, et al. Multi-scale climate variability of the South China Sea monsoon: A review. Dynamics of Atmospheres and Oceans, 2009, 47(1-3): 15-37.*

   *A3: We acknowledge the reviewer's concern regarding the selection of spatial domains. We have now provided a clearer explanation for these choices and added relevant references (Gill 1980; Wang et al. 2009). Lines 137–142 and 157–162.*

4. *The authors only analyzed changes over the period 1991–2020, which is rather short. Why not consider a longer time span, for example starting from 1979?*

   *A4: We agree with the reviewer that a longer period would be ideal. However, due to data reliability, quality, and availability, we restricted our analysis to recent climatology (1991–2020). We have clarified this in the revised manuscript and also emphasized that future work will extend to longer historical and future projections as data availability improves.*

5. *Figure 2: Although the timing of the seasonal transition is generally consistent, a closer look reveals that different variables are not perfectly synchronized. What causes this asynchrony, and how might it affect the CPM index?*

   *A5: We agree with the reviewer's observation. The asynchrony among variables reflects expected Gill-type dynamical adjustments versus thermodynamic and radiative memory processes (Gill, 1980; Wang et al., 2009). In our CPM index, we account for this by averaging across the five standardized change dates and reporting their spread as uncertainty, providing a stable and physically grounded estimate of onset and withdrawal. This clarification has been added to the revised text Line 216-222*

6. *Does the monsoon transition timing (onset and withdrawal) defined in this manuscript show any significant correlation with the time series of surrounding regions and other*

*monsoon subsystems? For example, many studies have indicated that the monsoon onset over the Bay of Bengal is significantly correlated with that over the South China Sea and India (Xing et al. 2016; Hu et al. 2022). If such calculations are difficult, the manuscript should at least include some discussion on this aspect.*

*Xing N, Li J, Wang L. Effect of the early and late onset of summer monsoon over the Bay of Bengal on Asian precipitation in May. Climate Dynamics, 2016, 47(5): 1961-1970.*

*Hu P, Chen W, Chen S, et al. The leading mode and factors for coherent variations among the subsystems of tropical Asian summer monsoon onset. Journal of Climate, 2022, 35(5): 1597-1612.*

*A6: We thank the reviewer for raising this important point. In response, we have expanded the discussion and incorporated the suggested references (Xing et al., 2016; Hu et al., 2022). A supplementary figure (Figure S-6) has also been added to illustrate related aspects (Lines 223–229). Fig S-6*

7. *Some studies have already focused on the interdecadal delay of the South China Sea summer monsoon withdrawal in recent years. Is this delay related to the summer monsoon withdrawal over the Indochina Peninsula?*

*Wang X, Zhou W. Interdecadal variation of the monsoon trough and its relationship with tropical cyclone genesis over the South China Sea and Philippine Sea around the mid-2000s. Climate Dynamics, 2024, 62(5): 3743-3762.*

*A7: We appreciate the reviewer's valuable suggestion. We have now discussed the potential link between the interdecadal delay of the South China Sea summer monsoon withdrawal and the Indochina Peninsula, with reference to Wang & Zhou (2024). Relevant sentences and citations have been added (Lines 223–224).*